# Exploring New Mechanism of Depression from the Effects of Virus on Nerve Cells

**DOI:** 10.3390/cells12131767

**Published:** 2023-07-03

**Authors:** Xinxin Yu, Shihao Wang, Wenzheng Wu, Hongyuan Chang, Pufan Shan, Lin Yang, Wenjie Zhang, Xiaoyu Wang

**Affiliations:** 1College of Pharmacy, Shandong University of Traditional Chinese Medicine, Jinan 250355, China; yuxinxin030303@163.com (X.Y.); 15153967019@163.com (W.W.); 2College of Chinese Medicine, Shandong University of Traditional Chinese Medicine, Jinan 250355, China; wsh20020302@163.com (S.W.); 18764427511@163.com (H.C.); z18353257688@163.com (W.Z.); 3College of Acupuncture and Tuina, Shandong University of Traditional Chinese Medicine, Jinan 250355, China; beilun314@126.com; 4College of Nursing, Shandong University of Traditional Chinese Medicine, Jinan 250355, China; 19819170338@163.com

**Keywords:** depression, virus, microglia, astrocytes, oligodendrocytes

## Abstract

Depression is a common neuropsychiatric disorder with long-term recurrent depressed mood, pain and despair, pessimism and anxiety, and even suicidal tendencies as the main symptoms. Depression usually induces or aggravates the development of other related diseases, such as sleep disorders and endocrine disorders. In today’s society, the incidence of depression is increasing worldwide, and its pathogenesis is complex and generally believed to be related to genetic, psychological, environmental, and biological factors. Current studies have shown the key role of glial cells in the development of depression, and it is noteworthy that some recent evidence suggests that the development of depression may be closely related to viral infections, such as SARS-CoV-2, BoDV-1, ZIKV, HIV, and HHV6, which infect the organism and cause some degree of glial cells, such as astrocytes, oligodendrocytes, and microglia. This can affect the transmission of related proteins, neurotransmitters, and cytokines, which in turn leads to neuroinflammation and depression. Based on the close relationship between viruses and depression, this paper provides an in-depth analysis of the new mechanism of virus-induced depression, which is expected to provide a new perspective on the mechanism of depression and a new idea for the diagnosis of depression in the future.

## 1. Introduction

Depression, or Major Depressive Disorder (MDD), is a major public health problem that affects both individuals and society as a whole [1]. The World Health Organization ranks depression as the fourth most common ailment. The prevalence of depression most commonly impacts younger ages, and the affected groups are now involved in colleges and primary and secondary schools. Consequently, investigating the causes of depression is of the utmost importance [2].

Depression is mainly characterized by emotional sadness, slow thinking, slow movements, less voluntary activity, cognitive problems, and physical symptoms such as difficulty sleeping and fatigue. Depression has a long incubation period and recurrent episodes [3]. Some studies have shown that depressed patients have more negative cognitive biases than those not suffering from the disorder and more often process information in many negative ways, such as making arbitrary inferences and overstating them [4]. As a common mental disorder related to suicidal intention, the etiology of depression is unclear, but the current understanding is that many factors, such as biological, psychological, and social environmental factors, are involved in the pathogenesis of depression. Researchers also tend to focus on the interaction between genetic and environmental or stress factors and the effects of this interaction on the development of depression [5].

At present, the mechanism of depression is not clear, but many studies have shown that it is closely related to glial cells [6]. In addition, recent research found that some viral infections are inextricably linked to the development of depression. In this paper, we review and combine the relevant literature to demonstrate this connection. We used severe acute respiratory syndrome Coronavirus 2 (SARS-CoV-2), Borna disease virus 1 (BoDV-1), human immunodeficiency virus (HIV), Zika virus (ZIKV), and human herpes virus 6 (HHV-6) to study the relationship between viruses and depression. We found that all five viruses can infect and damage astrocytes, oligodendrocytes, and microglia and that damage or changes to all three types of cells can lead to depression. Therefore, we hypothesized that these five viruses are capable of damaging the central nervous system or causing neuroinflammation, ultimately leading to the development of depression by causing damage to three different types of glial cells. In addition, we found that viral infection can directly influence the onset and course of depression in other ways, such as promoting the release of pro-inflammatory factors and regulating hormone release and protein expression. This study’s findings have the potential to facilitate further investigations into the causes, treatments, and prevention of depression.

## 2. The Classical Hypotheses of the Pathogenesis of Depression

The classical hypotheses for the pathogenesis of depression include neurotransmitters, the HPA axis, neuroplasticity, and neuroimmunity. Related studies have shown that cytokines can affect the metabolism and function of neurotransmitter systems such as serotonin (5 HT), norepinephrine (NE), dopamine, and glutamate; these effects may underly the pathophysiology of inflammation-induced depression [7,8]. In some depressed patients, the subcallosal cingulate cortex (SCC), also known as Broadman’s area 25 (BA25) or subgenus cingulate (Cg25), is abnormally active. Indeed, deep-brain stimulation located in area 25 of the nerve can reduce depressive symptoms in patients with abnormal neural activity [9]. Moreover, the release of inflammatory factors can alter neuroendocrine function, neurotransmitter systems, and neuroplasticity, which can, in turn, contribute to the course of depression. It was shown that increased IL-6 concentrations are significantly associated with hypothalamic–pituitary–adrenal (HPA) axis disorders and that depressed patients usually show elevated cortisol. In depression, the hyperactivation of the HPA axis leads to glucocorticoid receptor feedback insensitivity and the overproduction of other pro-adrenocorticotropic secretory hormones [10]. In addition, chronic promethazine (antidepressant) treatment significantly downregulates the plasma levels of the adrenocorticotropic hormone and corticosterone [11], suggesting that the HPA axis may be an important regulatory target in depression. In addition, brain changes associated with major depression have been reported in different areas, such as the hippocampus, amygdala, caudate nucleus, and nucleus accumbens [12]. In addition, the onset of depression is accompanied by inflammation, which also makes the body’s immune functions change, thereby affecting the normal physiological function of neuroimmunity, which is another important cause of depression [13].

Nerve cells can play various roles in the pathogenesis of depression. For example, neurons can regulate the release of neurotransmitters, and glial cells can both release neurotransmitters and activate neurons [14]. In addition, the microglia in glial cells, as neuroimmune effector cells, can affect neuroimmunity and release mediators after activation, stimulating the HPA axis and leading to dysregulation of the HPA axis. Microglia also play an important role in the destruction of neuroplasticity, which leads to the pathogenesis of depression [15,16]. Neuronal plasticity is another important factor in depression that affects neuroimmunity, thus further affecting depression. All of these factors suggest that nerve cells play an important role in the pathogenesis of depression.

## 3. The Role of Neurons and Glial Cells in Depression

### 3.1. The Association between Neuronal Cells and Depression

Neurons, or neuronal cells, are the most basic structural and functional units of the nervous system and are divided into two parts: the cell body and the protrusion. In recent years, the mechanisms by which neurons regulate relevant physiological functions and thus produce depression have been elucidated.

Extensive studies have shown that inflammation negatively affects mitochondrial health, leading to excitotoxicity; oxidative stress; energy deficiency; and, ultimately, neuronal death. In addition, damaged mitochondria can release multiple molecular patterns associated with damage, which can lead to a cycle including oxidative stress, mitochondrial damage, etc. This vicious cycle can be involved in regulating the mechanisms of inflammation-related depression, indicating that inflammation-induced neuronal death may be an important factor in the pathogenesis of depression [17]. Related studies suggested that although impaired dopaminergic neurotransmission is not considered a core neurochemical alteration in depression [18,19], there is now significant evidence that inflammation preferentially affects midbrain dopaminergic neurons such as by reducing dopamine synthesis and release and increasing dopamine reuptake [20]. Related studies have found that the antidepressant effects of amantadine may be due to an increase in extracellular DA concentration in the striatum and/or the indirect neuroprotective effect on dopaminergic neurons in the substantia nigra. Another possible cause of depression is a decrease in the reuptake and release of dopaminergic neurons [21]. Moreover, it was experimentally demonstrated that glutamate and hippocampal neuronal apoptosis are, respectively, key signals for and direct contributors to diabetes-associated depression. A previous study further suggested that in diabetes-related depression, the abnormal Glu–GluR2–Parkin pathway leads to the mitochondrial autophagy-mediated apoptosis of hippocampal neurons, showing that depression is also inextricably linked to hippocampal neurons [22]. Based on the above studies, the connection between neurons and depression is mediated mainly through inflammation and associated with apoptosis. Thus, the various responses mediated by neurons may be key to explaining the pathogenesis of depression in the future.

### 3.2. The Association between Glial Cells and Depression

Related studies have found that some triggers of depression are associated with glial cells [23]. This association is believed to be closely related to many nervous system diseases [24]. Taking astrocytes, oligodendrocytes, and microglia as examples, we analyze the association between these three types of glial cells and depression.

#### 3.2.1. Astrocytes and Depression

Astrocytes are a highly heterogeneous population of nerve cells responsible for central nervous system homeostasis, contributing to central nervous system homeostasis and providing defense against a variety of harmful effects. For example, astrocytes have the potential to promote or prevent inflammation and neurodegeneration by responding to signals in the microenvironment [25]. Chronic low-grade inflammation may lead to changes in brain structure and synaptic plasticity, leading to neurodegeneration, coupled with decreased neuroprotection and neuronal repairs due to increased glucocorticoid levels. These symptoms may be the initial pathological markers of depression and a prelude to dementia. Therefore, maintaining the normal physiological state of astrocytes is of great significance for the prevention and treatment of depression [26,27]. The dysfunction of the purinergic system in astrocytes is a typical example of this phenomenon. ATP released from astrocytes can regulate depression-like behavior in animal models and may also regulate clinical depression in patients. Astrocytes have purinergic receptors, such as adenosine A2A receptors and P2 × 7 and P2Y11 receptors. These receptors further regulate depression mechanisms by mediating neuroinflammation, neuroglial transmission, and synaptic plasticity in depression-related regions (e.g., the medial prefrontal cortex, hippocampus, and amygdala) [28]. This mechanism further shows that the relationship between depression and astrocytes is inseparable and that astrocyte damage can lead depressed patients into a vicious cycle of increased symptoms and astrocyte damage.

Astrocytes express a variety of neurotransmitter receptors, including the serotonin 5-HT2B receptor, and interact with neurons in the synapses. In major depression, the number, morphology, and functions of astrocytes deteriorate, which can lead to neurotransmitter imbalance and abnormal synaptic connections, exacerbating depression [28]. In addition, studies have found that a positive astrocyte glial fibrillary acidic protein immune response is associated with suicide in depression and speculated that the density of astrocyte IR–vimentin and GFAP–IR astrocytes in brain tissue will change when depression occurs, decreasing the number of astrocyte primary processes. The above factor further indicates that depression is closely related to astrocytes. Damage to the astrocytes will cause depression patients to fall into a vicious cycle of worsening symptoms and astrocyte damage [29]. In addition, relevant studies have shown that the KIR6.1-K-ATP channel (Kir6.1/K-ATP), as a metabolic stress receptor, is significantly expressed in astrocytes. Kir6.1 interacts with NLRP3 to prevent the assembly and activation of the NLRP3 inflammasome, thus inhibiting the programmed death of astrocytes. This activity may provide an important direction for the treatment of depression by regulating astrocytes [30]. Thus, astrocytes may be involved in the production of depression through a variety of mechanisms, including mediating neuroinflammation and metabolic dysfunction and leading to potassium-channel-driven neuronal explosion [31]. The involvement of astrocytes in the pathogenesis of depression is currently understood primarily in neuroinflammatory and metabolic responses and could be elucidated, in the future, at other mediating levels.

#### 3.2.2. Oligodendrocytes and Depression

Oligodendrocytes are myelin cells of the central nervous system that are differentiated from oligodendrocyte precursor cells (OPCs) under the regulation of various factors and can generate an insulating myelin sheath to promote the rapid conduction of axon action potential. OPCs can express a variety of neurotransmitter receptors and ion channels to maintain cell ion and water homeostasis and metabolism. In this way, the integrity of neurons and axons can be maintained, whereas the destruction of the integrity of neurons and axons can accelerate the progression of depression [32,33]. Studies have shown that oligodendrocyte lineage cells, including mature oligodendrocyte (OLs) and oligodendrocyte progenitor cells, have many important CNS-related functions, such as forming myelin sheaths that wrap axons of the central nervous system, mediating some forms of neuroplasticity, expressing neurotransmitter receptors, and enabling communication with neighboring neurons and axons. Such cells also provide nutritional and metabolic support for axons. Consequently, OLs in patients with depression will continue to change, further demonstrating the close relationship between oligodendrocytes and depression [34]. In addition, studies have shown that oligodendrocytes can be directly coupled to astrocytes in the neocortex [35] and that the metabolism of astrocytes, neurons, and oligodendrocytes will interact with each other. Therefore, we speculate that astrocytes and oligodendrocytes will affect each other’s metabolism through mutual influence and further regulate the onset and aggravation of depression [36]. Based on the above studies, astrocytes and oligodendrocytes can influence together through a synergistic relationship in the pathogenesis of depression, providing new ideas for the interpretation of this pathogenesis in the future.

#### 3.2.3. Microglia and Depression

Microglial cells are tissue-specific macrophages in the central nervous system that play an important role in neuroinflammation. During the generation of neuroinflammation, some microglial cells are activated and transformed into pro-inflammatory (M1) or anti-inflammatory (M2) phenotypes. Increasingly more studies have found that an increase in the microglia fine M1 type can promote the onset of depression. This phenomenon is also consistent with the condition that depression develops through an inflammatory pathway. In the central nervous system, neurons regulate microglial cells according to their own state and subsequently regulate the activity of neurons, which is reflected in the observation that some soluble factors are mainly expressed in neurons, and their receptors are mainly expressed in microglia. For example, CX3CL1 is highly expressed in neurons. Its receptor CXCR1 is highly expressed in the microglia. Soluble factors released by the microglia can also affect neuronal activity and the transport of neurotransmitter receptors. For example, TNF-α released by activated microglia can regulate synaptic plasticity through the release of glutamic acid. In addition, the activation of PI3K/Akt, ERK1/2, and mitogen-activated protein kinase (MAPK) is associated with neuroinflammation and the activation of M1 microglia. Notably, the activation of the PI3K/Akt pathway is critical for neurite repair and anti-apoptosis in central nervous system injury. These mechanisms all confirm that microglia influence the course of depression by regulating inflammation and neuronal activity. Moreover, the activation of adenosine 5’-monophosphate-activated protein kinase (AMPK) is also related to the activation of microglial cells, which can induce antidepressant effects by enhancing hippocampal neurogenesis and PKC-signaling pathways in neurons [37]. In addition, as a very important resident immune cell in the brain, microglial cells have an immune function in the brain and can further affect depression by regulating inflammation, synaptic plasticity, and the formation of neural networks. MDD is a mental disease that affects different cell dysfunctions in the brain and is associated with inflammation. Therefore, patients with depression will fall into a vicious cycle where pathological changes in microglial cells and the aggravation of MDD promote each other [38].

In addition, microglia can usually solve emerging immune and psychological problems by regulating the activity between the nerves and the immune system. For example, microglia can respond to neuroinflammation caused by stress and regulate neurons and astrocytes by releasing pro-inflammatory cytokines and their metabolites, thus regulating depression. This phenomenon is related to the expression degrees of cytokines and a variety of external environmental conditions. Therefore, microglia in patients with depression are generally abnormal, which deregulates the depressive mood of patients and aggravates their condition [16]. All the abovementioned microglial cells can form central nervous system inflammation via phagocytosis, clearing apoptotic cells, releasing inflammatory cytokines, etc., so their dysfunction is associated with a variety of neurological diseases, including depression. These diseases are also called “microglia disease” [39].

Currently, studies suggest that all molecular pathways leading to MDD may be linked to neuroinflammation and hippocampal degeneration through microglia-related neuroinflammation. Microglia can destroy neuroplasticity, affect neuroprotection, and overexpress cellular inflammatory factors, leading to the deterioration of neuroinflammation and depression. This result also suggests that microglia can have neuroprotective and neurotoxic effects that depend on factors such as the expression of cytokines and aging, the presence of pathogens and stress proteins, and external environmental conditions [16].

To summarize, astrocytes, oligodendrocytes, and microglia all play a defensive role in the occurrence of depression under normal physiological conditions. When these components experience pathological changes, they promote neuroinflammation by releasing inflammatory factors or induce depression by disrupting immune responses, the synaptic transmission of neurons, or neuronal integrity and other mechanisms. This outcome suggests that these three types of glial cells are closely related to depression [40].

Neurons and glial cells, as important components of the CNS, were shown to play an important regulatory role in the development of depression. For the regulation of neurons and depression, most current studies focus on dopamine neurons and hippocampal neurons [21,22]. Glial cells also have an irreplaceable role in the development of depression due to their different physiological functions. A clear analysis of the mechanism of CNS and depression would help summarize the mechanisms underlying the viral production of depression. Moreover, according to the above analysis, we learned that glial cells could affect depression by regulating various avenues of interaction, such as hormone release and metabolism, and by regulating neurons [29,37]. Therefore, we chose glial cells as the research object to explore part of the mechanism through which a virus can affect depression.

Table 1 was developed based on the discovery that glial cells have the potential to play a role in the regulatory processes associated with depression. We hope that this table will make it easier to understand the role that the various mechanisms play in this process.

## 4. Analysis of the Mechanisms by Viruses That Affect Depression

In recent years, many researchers have shifted the prospective cause of depression to viruses. A meta-analysis found that depression is associated with Borna disease virus, HSV-1, varicella zoster, and Epstein–Barr virus [41]. Related studies have shown that viruses produce depression specifically by affecting glial cells [42]. Viruses can infect glial cells in different ways, and glial cells can produce immune responses that damage the brain and lead to depression.

### 4.1. SARS-CoV-2

#### 4.1.1. Overview of SARS-CoV-2

According to the findings of one study, infection with SARS-CoV-2 may be a cause of MDD [43]. In late December 2019, an epidemic of viral pneumonia caused by SARS-CoV-2, a novel coronavirus, was reported in Wuhan, China. The pneumonia produced by SARS-CoV-2 was subsequently dubbed Coronavirus disease 2019 (COVID-19) [44]. The most important clinical manifestation of SARS-CoV-2 is severe pneumonia [45]. COVID-19 is primarily a respiratory disease, but it has many neurological complications that adversely affect the nervous system, primarily through cerebrovascular disease and immune-mediated neurological disorders [46]. Some studies have shown that SARS-CoV-2 can invade the nerves through retrograde axonal transport when it is transmitted from mother to child, resulting in a neurodevelopmental disorder and affecting the normal development of the nerves [47].

#### 4.1.2. Analysis of the Mechanism of SARS-CoV-2 Affecting Depression

In a clinical study, patients infected with SARS-CoV-2 showed a tendency to develop depression, and patients aged 18 to 50 years and male patients aged 65 years and older were more likely to have depression [48]. Related studies have shown that SARS-CoV-2 infection can increase brain inflammation, leading to increased oxidative stress and mitochondrial damage [49]. It was suggested that mitochondrial dysfunction is also involved in the disease development of depression [50,51]. Thus, SARS-CoV-2 can cause depression by producing mitochondrial damage. Clinical observations indicate that many patients with SARS-CoV-2 suffer from malnutrition. Some symptoms, such as dyspnea, hyposmia, anorexia, dysphagia, nausea, vomiting, and diarrhea, may lead to weight loss and malnutrition. SARS-CoV-2 can invade the oral mucosal epithelium, cause painful oral lesions and mouth ulcers, and significantly reduce nutrition in patients with COVID-19. In addition, SARS-CoV-2 increases patients’ anxiety, reduces their appetite, and exacerbates malnutrition [52,53]. COVID-19-induced malnutrition affects both peripheral and central serotonergic pathways through tryptophan (TRP) deficiency. Tryptophan is used to synthesize brain serotonin, which reduces depressive symptoms. Previous studies have shown the effects of malnutrition on serotonin and depression [54,55]. Based on these results, SARS-CoV-2 can cause depression by damaging mitochondria or causing malnutrition.

It was discovered that many neurological-like complications caused by SARS-CoV-2 are inextricably connected to the virus’s effect on glial cells [56]. In addition, many patients show a variety of neurological symptoms, including a loss of smell, nausea, dizziness, encephalopathy, and stroke, and there is a high prevalence of inflammatory central nervous system syndrome [57]. It was also shown that SARS-CoV-2 could induce depression by assaulting the central nervous system through the direct invasion of CNS neurons, glial cells, and proliferation, which results in apoptotic neuronal and glial cell destruction [58]. The role of astrocytes in the pathology of SARS-CoV-2 involves aging, neurodegenerative diseases, and environmental factors. Major depression may also be caused by decreased ATP release from brain astrocytes [23]. Depression is thought to be caused by glial synaptic dysfunction, which forms astrocyte synaptic circuits by releasing glial transmitters such as ATP. Microglia, the central nervous system’s resident macrophages, are also thought to be involved in this process. The medial prefrontal cortex (PFC) and hippocampus are important related structures, and there is evidence that low local ATP concentrations may lead to the development of P2 × 7Rs on microglia/glial cells in related brain regions, causing depression [59]. SARS-CoV-2 causes an activation of the microglia, which is closely related to the brain damage suffered by patients with SARS-CoV-2 [60]. Related studies have shown that microglia in patients with SARS-CoV-2 release more cytokines into the central nervous system [61]. Among proinflammatory cytokines, IL-6 is elevated during SARS-CoV-2 infection and plays an important role in the pathogenesis of depression in COVID-19 patients. This study suggests that IL-6 concentration may be directly related to the severity of depression in infected patients [62]. Thus, SARS-CoV-2 can cause the onset of depression by damaging astrocytes and microglia. In addition, it was shown that SARS-CoV-2 is also responsible for the breakdown of myelin oligodendrocyte glycoprotein antibodies, which, in turn, has an impact on the central nervous system. It was also shown that some patients develop myelin oligodendrocyte glycoprotein antibody illness after receiving a COVID-19 vaccine [63,64]. In a clinical study, the generation of myelin oligodendrocyte glycoprotein disorders was shown to predispose patients to depression and contribute to its progression [65].

In conclusion, SARS-CoV-2 can damage the central nervous system and produce neuroinflammation by destroying the glial cells of neurons, thereby affecting the normal physiological functions of the central nervous system and causing the onset of depression.

### 4.2. Borna Disease Virus 1

#### 4.2.1. Overview of Borna Disease Virus 1

Borna disease virus 1 (BoDV-1; species Mammalian 1 orthobornavirus) can cause progressive meningoencephalitis, which occurs mainly in horses and sheep. Recent studies have shown that after receiving an organ transplant, BoDV-1 patients have symptoms of viral encephalitis [66]. It was proposed that BoDV-1 may be a neurogenic virus. At the same time, studies have shown that BoDV-1 preferentially infects the limbic system of the brain and forms persistent infections [67]. Epidemiological studies have shown that BoDV-1 infection exists worldwide and has a wide range of epidemics. Additionally, patients infected with BoDV also develop symptoms related to encephalitis [68]. During chronic BoDV-1 infection in animals, the serum and cerebrospinal fluid of many animals are affected, including naturally infected horses and experimentally infected rabbits, mice, and chickens [69,70,71]. These results suggest that BoDV-1 can infect both humans and animals and cause viral encephalitis.

#### 4.2.2. Analysis of the Mechanism of Borna Disease Virus Affecting Depression

Borna disease virus 1 (BoDV-1) differs from other neurotropic viruses in that it can persistently infect neurons in the central nervous system (CNS) without causing systemic cell death. Possibly due to its effect on the protein kinase-C-signaling pathway (PKC), the enhanced activity of activity-dependent neuronal networks is disturbed after BoDV-1 infection. Previous studies have shown that PKC and its associated signaling pathways are closely related to mental processes in the brain and the pathogenesis of mental disorders, including depression. The level of PKC was found to be decreased in depressed rats, while the treatment of depression with paroxetine can increase the level of PKC in rats, indicating that PKC content can represent the development of depression [72]. Related studies have shown that BoDV-1-infected cells are enriched in the structural protein BoDV-P, which causes massive phosphorylation of PKC, leading to a decrease in normal PKC levels, potentially causing depression [73]. Related studies have shown increased astrocyte and neuronal cell death in BoDV-1-infected vivo, suggesting that BoDV-1-P can directly affect astrocyte functions. Elevated levels of BoDV-P expression can also cause brain damage and behavioral impairment in mice [74]. Previous studies have shown that BoDV infection leads to the inhibition of synaptic enhancement and rapid axonal transport. These phenomena can promote the secretion of BoDV-1-derived phosphoprotein P, which acts as a competitive inhibitor of PKC and can reduce PKC levels, leading to depression [75].

In past studies, researchers found antibodies against BoDV-1 with BoDV-1 transcripts in the blood of animals and human psychiatric patients infected with BoDV-1. In addition, infectious strains of BoDV-1 were isolated in two patients with bidirectional acute depression and one patient with chronic obsessive-compulsive disorder [76]. BoDV-1 RNA and antigens were detected in brain tissue when researchers performed autopsies on schizophrenic patients. Additionally, mild inflammatory changes in the hippocampus were found in patients in the early stages of BoDV-1 infection [77]. Meanwhile, in rats chronically infected with BoDV-1 since birth, only subtle and transient inflammatory changes were seen in the brain tissue, despite the animals exhibiting neurobehavioral and neurochemical abnormalities [78]. It was suggested that infected neurons might be damaged by T-cell-mediated cytotoxicity or die due to the excessive release of inflammatory cytokines from the microglia or glutamatergic storms. The underlying cause of neuronal damage is the failure of infected astrocytes to regulate brain glutamate levels [79]. The kainate 1 (KA-1) neurotransmitter receptor of the glutamate receptor family is not expressed in CA1 neurons but is present in CA3 neurons. In contrast, available studies suggest that viral RNA is present in CA3 neurons but not in CA1 neurons, indicating that KA-1 may be a key receptor in controlling BoDV-1 [80]. A study of depressed patients with BoDV-1 infection reported a reduction in both depressive symptoms and BoDV-1 infection after treatment with the antiviral drug amantadine [81]. Additionally, in a double-blind placebo-controlled phase II RCT design that cross-linked depression and BoDV-1 infection to assess the antidepressant and antiviral effects of amantadine, amantadine was found to be effective in reducing viral levels in patients and improving symptoms associated with depression [82].

BoDV-1 replicates in the nucleus and acts mainly on hippocampal neurons but can also act on astrocytes and oligodendrocytes in the brain. The study noted that BoDV-1 infection in neonatal rats produces persistent CNS infection with a range of neurodevelopmental abnormalities and complex behavioral changes similar to those seen in autism, schizophrenia, and depression [83]. It was shown that human wild-type BoDV-1 (Hu-H1) isolated from depressed patients differs significantly in terms of its in vitro biological properties (human OL cells) in a laboratory-adapted strain (V strain) isolated from diseased horses. Hu-H1 promotes apoptosis in oligodendrocytes, whereas the V strain inhibits apoptosis in oligodendrocytes [84]. In the mammalian central nervous system, glutamate and aspartate serve as the main excitatory neurotransmitters for the N-methyl-D-aspartate (NMDA) receptor, and in related metabolomic studies, it was found that reduced levels of glutamate and aspartate lead to decreased excitatory levels of NMDA. Reduced glutamate in the brain involves the functional impairment of the glutamatergic system, which causes damage to astrocytes and can lead to enhanced neuronal toxicity [85]. The presence and differences between human and animal BoDV-1 strains were highlighted in past studies. Furthermore, it was found that Borna virus infection could disrupt the metabolic profiles of several metabolites in human oligodendrocytes cultured in vitro. Among them, a significant increase in mitogen-activated protein kinase (Ras/Raf/MEK/ERK)-signaling cascade occurred in astrocytes and oligodendrocytes infected with BoDV-1 [86,87]. Related studies suggest that RSKL 2, as a downstream substrate of ERK/RSK, may help increase the kinase cascade in the Ras–Raf–MAPK signaling pathway, thereby affecting the transcription and expression of downstream genes and promoting the production of neurotrophic factors. RSKL 2 may also play a role in promoting neurotrophic factors, cell division, proliferation, and neuroplasticity and ultimately mediate the outcome of antidepressant treatment. This result indicates that Ras/Raf/MEK/ERK can influence the treatment of depression and participate in the disease process [88].

In conclusion, the Borna virus has the potential to have wide-ranging impacts on the human brain and neurological functioning. One of the ways in which the virus can have such an impact is by causing damage to the three aforementioned glial cells, which can then result in the development of depression.

### 4.3. Human Immunodeficiency Virus

#### 4.3.1. Overview of Human Immunodeficiency Virus

The HIV epidemic is caused by zoonotic infection with the monkey immune defense virus in African primates [89]. A key factor in the worsening of illness on a worldwide scale is the infection and spread of the human immunodeficiency virus [90]. AIDS, also known as acquired immunodeficiency syndrome, is caused by infection with the human immunodeficiency virus (HIV) [91]. The transmission of this virus occurs most commonly through sexual contact, blood contact, and mother-to-child contact. Infection with this virus can result in severe immunodeficiency [92]. Some studies have shown that HIV glycoprotein Gp120 can damage rapid axonal transport by activating the Tak1-signaling pathway, which can cause damage to DRG neurons and affect the normal physiological activities of the nerve [93].

#### 4.3.2. Analysis of the Mechanism of Human Immunodeficiency Virus Affecting Depression

The most common neurological disorder in HIV patients is depression. Studies have shown that the prevalence of depression among HIV-infected individuals is three times higher than that of the general population. The chronic activation of inflammatory mechanisms that disrupt CNS functions may contribute to this outcome. Studies have documented a significant relationship between depression and increased proinflammatory cytokines, such as IL-6, IL-1β, TNF-α, and CRP [94]. The continuous progression of chronic inflammation will lead to a continuous increase of inflammatory factors in the central nervous systems of patients and then deepen the effects of depression. Furthermore, depression severity has been associated with decreased BDNF. Similarly, increased proinflammatory cytokines and decreased BDNF were associated with the progression of HIV and its associated neurocognitive disorders [95]. Parasympathetic nerves are stimulated by inflammatory factors, resulting in hypotonia, pain, psychomotor disorders, depression, and other sickness behaviors, which are mediated by the vagus nerve (VN). These are some of the most common symptoms of HIV with depression [96].

Although astrocytes are significantly resistant to infection by cell-free HIV in vitro, these cells are effectively infected by cell-to-cell contact, where immature HIV budding from lymphocytes through cell contact, with the ability to bind directly to CXCR4, triggers the fusion process in the absence of CD4 cells. The CCR5 coreceptor (CCR5) is one of the main co-receptors of HIV-1’s invasion of body cells, which promotes HIV replication in the human body. However, the CCR5 receptor has not been detected in astrocytes [97]. Some experiments showed that LPS stimulation could establish a depression model in mice, and esculin has a protective effect on LPS-induced depression. LPS stimulation may function by inhibiting the TLR4/nf-κB-signaling pathway regulated by CCR5, which indicates that CCR5 can be involved in the regulation of depression [98].

HIV infects astrocytes in a restrictive manner. The proliferation of this disease requires a high expression of Nef, a major virulence factor for HIV replication and disease progression. By introducing HIV molecular clones with intact Nef and HIV molecular clones without Nef (nonsense Nef mutants) into human primary astrocytes and comparing gene expression in astrocytes, one study revealed a negative regulatory role of intact Nef in HIV replication and astrocyte pathogenesis [99]. Experimental validation found that Nef expression led to increased glutamate uptake, and decreased glutamate release from astrocytes and increased astrocyte proliferation [100]. A decrease in glutamate can induce the formation of depression, so HIV may induce depression by mediating astrocytes to regulate glutamate levels.

According to several studies, HIV infection stimulates the microglia; causes the downregulation of glycogen synthase kinase-3β (GSK-3β), mitogen-activated protein kinase 3 (Mapk3), and other proteins; and leads to a significant increase in IL-1β and IL-6, which indicates that HIV-infected microglia can promote neuroinflammation [101]. GSK-3β can be involved in the regulation of a variety of mental disorders, such as bipolar disorder and major depression [102]. Recent studies have shown that GSK-3 can affect mitochondrial movement through effects involving schizophrenia 1 (DISC1) and microtubule stability, thereby disrupting mitochondrial trafficking to dendrites and axons through motor proteins [103]. Furthermore, HIV infection causes a massive decrease in oligodendrocytes but without a significant difference in the number of astrocytes and microglia, supporting the conclusion that HIV causes demyelination and axonal dysfunction. Additionally, it was hypothesized that the aforementioned phenomenon could further affect the central nervous system or lead to neuroinflammation by affecting glial cells, leading to depression [104]. A decrease in the number of oligodendrocyte could be interpreted as indicating increased cell death or damage during the proliferation, maturation, or differentiation of oligodendrocyte precursors. Accumulating evidence suggests that HIV viral proteins directly damage oligodendrocytes and that extensive demyelination is a feature of HIV-associated neurocognitive disorders and a major contributor to depressive manifestations [105].

Consequently, the relationship between HIV and neuroinflammation can be further elucidated by exploring the relationship between HIV and the number of different glial cells, which, in turn, illustrates the relationship between HIV and MDD.

### 4.4. Zika Virus

#### 4.4.1. Overview of Zika Virus

The Zika virus was initially detected in 1947 during research on the yellow fever virus in the Zika forest of Uganda and first isolated from a serum sample from a rhesus monkey. In recent years, the Zika virus has been linked to an increase in cases of birth defects in infants [106]. Zika virus infection is most commonly spread by the bite of an infected female Aedes aegypti or Aedes albopictus mosquito [107]. Infection with ZIKV is often painless and symptomless, but in rare instances, this virus can produce fever, rash, myalgia, headache, and nonsuppurative conjunctivitis [108]. In addition, ZIKV was associated with various neurological complications such as microcephaly and other birth defects, Guillain–Barre syndrome, meningoencephalitis, myelitis, and various ophthalmic abnormalities [109].

#### 4.4.2. Analysis of the Mechanism of Zika Virus Affecting Depression

Zika virus is a rapidly emerging yellow fever virus that is associated with a wide variety of congenital neurological manifestations, such as acute disseminated encephalomyelitis, myelitis, and cerebrovascular complications. Gene expression arrays of neural stem cell progenitors and differentiation markers indicate that infection with ZIKV reduces the number of neuronal and oligodendrocyte progenitors while increasing the number of astrocyte progenitors, further allowing astrocyte infection to increase the transcription of key genes involved in the antiviral response and affect neurodevelopment. After ZIKV infection, Sox2, Tuj1, NeuN, DAT, and synaptophysin were found to be down-regulated, while DCX and nestin were found to be up-regulated [110]. Previous studies have shown that the expression of Sox2 and DAT is inhibited in animal models of depression, while the expression of DCX is increased during the depression-like behavior induced by LPS [111,112]. The expression of the representative related genes could be involved in the development of depressive disorders. Astrocytes are particularly vulnerable to ZIKV infection and were hypothesized to function as a source of pro-inflammatory cytokines in brain tissue infected with ZIKV. The study showed that the expression of inflammatory proteins IL-6, 8, and 12 was increased in astrocytes infected with ZIKV [113]. This correlation between IL-6 and depression could indicate that the production of neuroinflammation after ZIKV infection promotes the development of depression.

ZIKV also infects glial precursor cells during brain development, meaning that ZIKV infection halts oligodendrocyte development by interfering with the proliferation and differentiation of oligodendrocyte precursor cells [114]. It was concluded that clinically relevant ZIKV isolates could directly affect oligodendrocytes based on observing a large number of apoptotic oligodendrocytes in the white matter of the spinal cord infected with BoDV-1, along with a restricted microglial cell response, including the expression of NLRP3 inflammatory vesicles [115]. Relevant studies have indicated that the various pathways involved in NLRP3 activation are the main targets of NLRP3 inhibitors in the treatment of depression [116]. It was suggested that blocking the effect of the NLRP3 inflammasome is critical for predicting the release of inflammatory cytokines by mediators leading to depression. ZIKV can also affect the microglia and further increase the secretion of TNF-α, IL-6, IL-1β, and iNOS, leading to neuroinflammation [117]. Based on the above analyses, it was hypothesized that Zika virus could cause depression through three pathways: the infection of infected astrocytes, a reduction in the number of oligodendrocytes, and the production of microglial cell responses.

In recent years, studies have shown that exercise during prenatal Zika virus infection can prevent the development of depression in mothers and pups. This study confirmed, from a behavioral perspective, that Zika virus infection can cause depression in animals. The molecular mechanisms underlying changes in depressive behavior include enhanced ionized calcium-binding adapter molecule 1 (IBA-1) and glial fibrillary acidic protein (GFAP), as well as the levels of brain-derived neurotrophic factor (BDNF) in the hippocampus of female and male pups [118]. BDNF is related to the maintenance of central nervous system integrity in patients with mood disorders such as anxiety and depression; a decrease in BDNF expression can promote the occurrence of such mental diseases [119]. These studies suggest that the Zika virus can directly affect normal behavior and produce depression in animals.

### 4.5. Human Herpes Virus 6

#### 4.5.1. Overview of Human Herpes Virus 6

Based on its molecular signature, HHV-6 was found in 1986 to be the oldest human herpes virus [120]. HHV-6 is the causative agent of one of the most common childhood diseases, rosacea, and adult latent infection rates exceed 90% [121]. HHV-6 is a neuroaffinity virus, and some research suggests a link to Alzheimer’s disease [122]. HHVs are neurotropic viruses that induce severe chronic neurological diseases, including PNS and CNS. HHV-6 reactivation is associated with many systemic clinical manifestations, including lung, kidney, heart, brain, and gastrointestinal tract disorders. HHV-6 can infect a variety of central nervous system cells in vitro and is associated with several neurological diseases, including encephalitis, seizures, chronic fatigue syndrome, medial temporal lobe epilepsy, Alzheimer’s disease, and multiple sclerosis [123].

#### 4.5.2. Analysis of the Mechanism of Human Herpes Virus 6 Affecting Depression

It was proposed that infection with the human herpes virus 6 may be associated with the development of depression [124]. According to the findings of one study, the human herpes virus 6 (HHV-6) may be responsible for depression because it damages astrocytes [125]. Data from a clinical study indicate the more frequent detection of HHV-6A late protein in the cerebellum of patients with MDD. At the same time, HHV-6A and HHV-6B DNA and protein contents were found to be higher in MDD patients than in the controls. These results indicate that HHV-6A and HHV-6B are more common in major depression [126]. HHV-6 can lie dormant in the central nervous system and other tissues. When this virus is activated, it can cause cognitive and behavioral impairments. In addition, a study based on the results of a comprehensive analysis of HHV-6A-infected HA1800 cells identified several genes that are associated with neurological disorders. In particular, seven genes associated with CNS disorders, CTSS, PTX3, CHI3L1, Mx1, CXCL16, BIRC3, and BST2, were found to be altered, which led to the development of related neurological disorders [125]. Relevant studies have shown that PTX3 could be detected 3.5 times more strongly in patients with depression compared to the rates of those in the control group, indicating that PTX3 is positively correlated with the onset of depression and can be used as a marker of depression [127]. Human herpesvirus infection affects the brain and interferes with the function of microglia, causing chronic viral infection with mild neuritis. The long-term persistence of HHV may help maintain the corresponding immune response and cause persistent chronic low-grade neuroinflammation, thereby inducing and accelerating the brain aging process [128]. Clinical data indicated that the degree of brain aging was more pronounced in patients with depression than in the controls. This phenomenon could be observed from the onset of the disease (<3/6 months) throughout the first 2 years of the disease. There was little difference in the degree of brain aging after 2 years [129]. These results suggest that brain aging may be an indicator of the development of depression. In addition, both HHV and novel coronaviruses are associated with antibodies against myelin oligodendrocyte glycoproteins. These phenomena suggest that human herpesviruses have factors in common with other viruses in the sense that both may contribute to depression by affecting astrocytes, oligodendrocytes, and microglia [130].

Relevant studies indicated that the expression levels of Varicella–Zoster virus responder cell frequency (VZV–RCF) in patients with depression were lower than those in the control group and were negatively correlated with the severity of depressive symptoms. The VZV–RCF levels in patients with depression who received antidepressant treatment were higher than those in patients with depression who did not receive treatment, indicating that the VZV–RCF levels in HHV patients can be used as a relevant indicator for the detection of depression [131]. Recent studies have used SITH-1 as a specific protein marker for HHV-6 infection. The experimental results indicate that the expression of SITH-1 increased in HHV-6-infected animal models. At the same time, part of the HPA axis was enhanced, and the experimental animals developed depression-related symptoms [132]. Based on this study, it can be assumed that the cause of depression caused by HHV-6 is regulated through HPA. Studies have shown that ATP binds to the membrane purinergic P2 receptor (P2R) of neurons and astrocytes, leading to an increase in intracellular Ca^2+^ to activate the concentration of GSK-3β, which can effectively promote HSV-1 replication [133]. At the same time, relevant clinical studies have shown that repeated infection with HSV-1 can greatly increase the probability of depression [134]. This result indicates that HSV-1 can promote the onset of depression.

In summary, HHV-6 can affect the secretion of a variety of cytokines by affecting glial cells and leading to the expression of related genes. This virus can affect the normal physiological functions of the central nervous system through neuroinflammation, resulting in the pathogenesis of depression.

Based on the abovementioned mechanisms related to viral effects on glial cells leading to depression, Figure 1 illustrates the corresponding regulatory relationships.

## 5. Conclusions

According to previously conducted research, neuroglial cell pathology represents a probable underlying cause of MDD [135]. The dysfunction of neuroglia, such as astrocytes, oligodendrocytes, and microglia, is an important cause of depression [6], while infection with SARS-CoV-2, Borna virus, human immunodeficiency virus, Zika virus, and human herpes virus 6 all contribute, to some extent, to the dysfunction of astrocytes, oligodendrocytes, and microglia, which, in turn, affect the central nervous system and can lead to the development of depression through neuroinflammation. In addition to this mechanism, it was hypothesized that the action of glial cells on the creation of depression is connected to the concept that depression is produced by 5-hydroxytryptamine and norepinephrine [136]. Drug studies have shown that antidepressants can resist the reduction of astrocytes, further indicating that astrocytes provide nutritional, structural, and metabolic support to neurons, which is closely linked to the prevention and treatment of depression [137]. Furthermore, antidepressant treatment can not only regulate abnormal neurotransmitter concentrations in the synaptic cleft but also reshape neuronal circuits by affecting glial cells [138]. Based on this activity, it is hypothesized that many viruses can cause depression by damaging glial cells. This conclusion may provide new ideas and approaches for the study and treatment of depression and provide insight into the viral hypothesis of depression. Studies have shown that significant changes in the total cerebrospinal fluid protein and immunoglobulin in depressed and neurological patients present similarities between the two groups. However, the researchers failed to analyze cerebrospinal fluid in these subjects (C.S.F.). The detection of antibodies and statistically negligible seropositivity refute the hypothesis of depression, but the results of depression patients and neurological patients leading to similar protein changes raised the possibility of other etiologies. The depression hypothesis faces significant controversy, indicating that we should deepen future research in this area to explore the link between the virus and depression.

A limitation of this article is that it covers only the active and passive impacts of glial cells on depression but does not thoroughly explore the mechanism of viral activity on glial cells. More research must be conducted on this aspect. Future studies should also determine whether the direct detection of viral levels in depressed individuals as an indication of the severity of the signal could represent a viable future avenue for therapeutically meaningful development.

A thorough study of the literature was used in this article to illustrate the connection between glial cells and depression. Moreover, this article discussed the influence that viruses have on depression, which may provide useful therapeutic ideas for the development of future treatments in this area.

## Figures and Tables

**Figure 1 cells-12-01767-f001:**
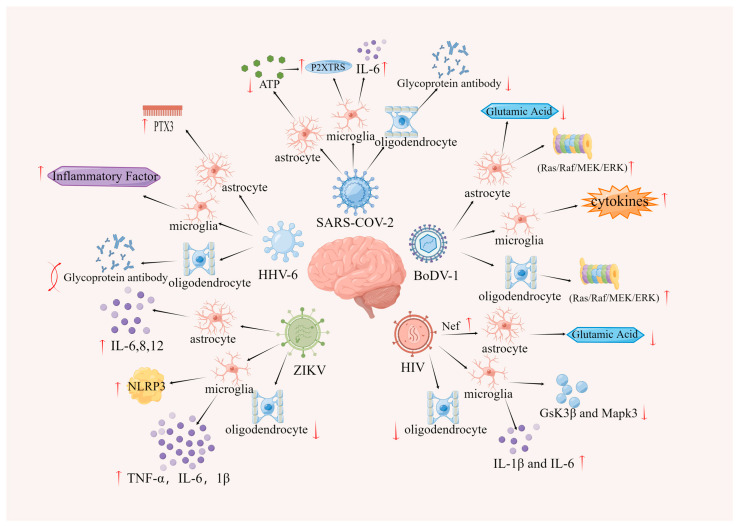
Mechanisms by which viruses affect depression through glial cells.

**Table 1 cells-12-01767-t001:** Mechanisms by glial cells affect depression.

Cell Type	Mechanism	Impact	References
Astrocytes	Release of ATP	Mediates neuroinflammation, neural (glial) transmission, and synaptic plasticity to further regulate depression mechanisms	[28]
	Deregulation-regulated purinergic signaling	Develops and aggravates depression	[28]
	Quantity reduction and degradation	Neurotransmission imbalance and abnormal synaptic connections	[28]
	Express multiple neurotransmitter receptors and interact with neurons at the synapses	Imbalanced neurotransmission and abnormal synaptic connections, aggravating depression	[28]
	The density of IR–vimentin and GFAP–IR astrocytes in brain tissue is altered	Falling into a vicious cycle of increased disease and astrocyte damage	[29]
	Regulation of Kir6.1-K-ATP channels	Falling into a vicious cycle of increased disease and astrocyte damage. Contribute to depression	[30]
	Mediate neuroinflammation and metabolic dysfunction	Contribute to depression	[31]
Oligodendrocytes	Form a myelin sheath that encapsulates the CNS axons	Contribute to depression	[34]
	Mediate some forms of neuroplasticity and provide nutritional and metabolic support to the axons	Contribute to depression	[34]
	Interact with astrocytes and neurons	Contribute to depression	[36]
Microglia	Some microglia will activate to become pro-inflammatory (M1) phenotypes	Reduce neuroinflammation and promote the progression of depression	[37]
	Microglia activate to release soluble factors	Affect neuronal activity and trafficking of neurotransmitter receptors, regulating depression	[37]
	Regulation of inflammation and synaptic and neural plasticity	Impact the course of depression	[38]

## Data Availability

Not applicable.

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
