# Peer review of "Exploring New Mechanism of Depression from the Effects of Virus on Nerve Cells"

_cells, 2023, doi:10.3390/cells12131767_

Round 1

Reviewer 1 Report

The authors performed an interesting hypothesis of how nerve cells can be affected by virus inducing depression. The idea of the damage produce by viruses in nerve system cells is intriguing.

However, there are several problems when performing and analyzing the information obtained. One problem with the article is that it is not known whether it is a narrative review article or a systematic review article. The terms searched for in the web of science are mentioned in a good way but they are mentioned within the sections of the article but only on one basis. In addition, several paragraphs do not have the references used for their writing. 

On the other hand, the auhtors make a pie chart to express in percentages the number of articles found on each cell line when it would be better using a flowchart and express how many were related to the terms and how many were not.

I think that if the authors finish doing the review in a systematic review format, including databases such as Pubmed and some other (remembering that at least three databases are recommended for a systematic review) it could help to understand the global context and the information in a more complete and analytical. The use of the PRISMA diagram would help a lot for the real analysis of the available information and that the authors used for the study.

As for the wording, it has several writing flaws and there are some phrases that are not fully understood.

Reviewer 2 Report

See attached file!

Sound quality!

Reviewer 3 Report

The review of Yu et al. highlights a developing feature of depression, i.e. that it may be closely related to viral infections, such as SARS-CoV-2, BDV, 24 ZIKV, HIV, and HHV6, which infect the organism. They summarize new mechanisms of virus-induced depression concerning the secretion of a variety of cytokines affecting glial cells and leading to the impairment of normal physiological function of the CNS through neuroinflammation, resulting in the pathogenesis of depression.

In this context, I would recommend an effort to also correlate the well-known retrograde axonal transport of viruses to the depression, together with the engagement of neurotropic viruses in fast axonal transport, which is also dependent on the oligodendrocytes, regarding the axonal mitochondrial health and distribution. In fact, I believe that the potential neurological effects of viruses at the CNS level cannot be treated without considering their entry mechanism through retrograde axonal transport, which is held responsible for the onset of various brain injuries as well as potentially of the depression.

I recommend a mild re-editing of the English and the check of some typos in the text.

Round 2

Reviewer 1 Report

The authors only added a couple of sentences in the last part of the section paragraphs as "summary", which is not bad and can help to round out the understanding of the information.

However, the observations made during the first review by me were not actually resolved. The pie charts do not reflect anything, since they mention that this is what information was found. And those articles were all helpful? How many of those did you use? Was the information collected relevant to each of the subtopics?

As for English, I again found sentences that are not fully understood. I suggest review by a native speaker.

Author Response

Response to Reviewer 1 Comments

We are very glad to receive your review comments. Thank you for your recognition of our work and providing such professional suggestions for revision. We have carefully revised the manuscript according to the suggestions.

Point 1: However, the observations made during the first review by me were not actually resolved. The pie charts do not reflect anything, since they mention that this is what information was found. And those articles were all helpful? How many of those did you use? Was the information collected relevant to each of the subtopics?

Response 1: According to Reviewer 1 suggestions, we revised the article again. Since the style of the article we wrote is a narrative review article, it mainly summarizes the current research progress. The purpose of the pie chart is to visually show the reporting rate of the relationship between different glial cells and depression. The three types of glial cells involved in the pie chart correspond to the subtopics, but we only describe and summarize the main references. However, as Reviewer suggestion, because only Web of Science database was analyzed, the data displayed in the graph was not comprehensive enough. After careful consideration, it was decided to delete the information related to the pie chart.

Point 2: As for English, I again found sentences that are not fully understood. I suggest review by a native speaker. 

Response 2: For the language revision, we have used the website recommended by the journal of Cells (https://www.mdpi.com/authors/english) to have the language touched up, and we will provide proof of the touch-ups in the attachment.

Reviewer 2 Report

Cells-2445826 revised manuscript

Comments to authors

The revised manuscript addressing the role of virus infections in major depressive disorder (MDD) at the cellular perspective of glial cells, has been considerably improved, according to reviewers’ requests. Several mismatches in references have been corrected and lacking references included, extending those requested. This was contributing to a thorough revision of text passages.

In terms of Borna disease virus, the whole chapter 3.2 has been re-written as requested. While paragraph 3.2.2 has been improved by including formerly lacking aspects, the introducing paragraph 3.2.1 is still suffering from several misleading wordings:

Lines 360-361

-        The transplant recipients got organs from one and the same donor whose BoDV-1 infection was unknown and sub-clinical. The subsequent development of fatal encephalitis (2 cases) and blindness (1 case) led to further investigations of brain autopsy samples which demonstrated that transplant transmitted BoDV-1 was the causative agent, while other viruses could be excluded (ref. 67). This finding showed for the first time that BoDV-1 could cause encephalitis in humans like know in animals, although very rare.

Lines 362-364

-        Not the human encephalitis cases but previously long known animal infections had undoubtedly shown that BoDV-1 was a neurotropic virus that preferentially and persistently infects the limbic system (ref. 68 and 70). “Neurogenic” is the wrong term.

Lines 364-366

-        Epidemiological studies on human BoDV-1 infection were mainly done using blood infection markers (antibodies and antigens in serum; RNA in blood cells). Those studies compared the infection prevalence in psychiatric patients and healthy donors and found a significantly higher prevalence in patients than controls in several countries, suggesting a worldwide BoDV-1 infection (new reference Bode et al 2020). Nowadays, also narrow endemic regions were postulated based solely on encephalitis cases and a zoonotic reservoir. Both opposite concepts are under debate, as well as the clinical spectrum of human BoDV-1 infection (psychiatric diseases vs. encephalitis only). Molecular analysis argued in favor of worldwide infection and a broad clinical spectrum (ref. 69).

Lines 369-370

-        BoDV-1 infections of humans as well as of a broad range of mammals are meanwhile non-contestable. Also, encephalitis is a rare outcome, and asymptomatic courses are known in most hosts (ref. 72; Bode et al. 2020).

Bode L, Xie P, Dietrich DE, Zaliunaite V, Ludwig H. Are human Borna disease virus 1 infections zoonotic and fatal? Lancet Infect Dis. 2020 Jun;20(6):650-651. doi: 10.1016/S1473-3099(20)30380-7.

Taxonomy

Please use the new virus name, BoDV-1, throughout the chapter/manuscript, and not only in paragraph 3.2.1. This term should be uniformly applied.

In conclusion, apart from an otherwise sufficiently improved revised version, a more differentiated wording of paragraph 3.2.1 is recommended considering above details.
